# Effects of Intermittent Pneumatic Compression on Lower Limb Lymphedema in Patients with Type 2 Diabetes Mellitus: A Pilot Randomized Controlled Trial

**DOI:** 10.3390/medicina57101018

**Published:** 2021-09-25

**Authors:** Alessandro de Sire, Maria Teresa Inzitari, Lucrezia Moggio, Monica Pinto, Giustino de Sire, Marta Supervia, Annalisa Petraroli, Mariangela Rubino, Delia Carbotti, Elena Succurro, Antonio Ammendolia, Francesco Andreozzi

**Affiliations:** 1Physical Medicine and Rehabilitation Unit, Department of Medical and Surgical Sciences, University of Catanzaro “Magna Graecia”, 88100 Catanzaro, Italy; inzitari@unicz.it (M.T.I.); petraroli.annalisa@virgilio.it (A.P.); ammendolia@unicz.it (A.A.); 2Rehabilitation Medicine Unit, Strategic Health Services Department, Istituto Nazionale Tumori-IRCCS-Fondazione G. Pascale, 80131 Napoli, Italy; monicapinto@iol.it; 3Medical Clinic “D.S.G.”, Caserta Local Health Service, 81100 Caserta, Italy; desiregiustino@libero.it; 4Gregorio Marañón General University Hospital, Gregorio Marañón Health Research Institute, Dr. Esquerdo 46, 28007 Madrid, Spain; msuperviapola@gmail.com; 5Mayo Clinic, 200 First St. SW, Rochester, MN 55905, USA; 6Internal Medicine Unit, Department of Medical and Surgical Sciences, University of Catanzaro “Magna Graecia”, 88100 Catanzaro, Italy; mariangela.rubino@unicz.it (M.R.); deliacarbotti@gmail.com (D.C.); succurro@unicz.it (E.S.); andreozzif@unicz.it (F.A.)

**Keywords:** lymphodrainage, intermittent pneumatic compression, manual lymphatic drainage, lymphoedema, rehabilitation

## Abstract

*Background and Objectives*: Diabetes mellitus type 2 (T2DM) is a chronic disease associated with fluid accumulation in the interstitial tissue. Manual lymphatic drainage (MLD) plays a role in reducing lymphoedema, like intermittent pneumatic compression (IPC). By the present pilot study, we aimed to evaluate the efficacy of a synergistic treatment with MLD and IPC in reducing lower limb lymphedema in T2DM patients. *Materials and Methods*: Adults with a clinical diagnosis of T2DM and lower limb lymphedema (stage II–IV) were recruited from July to December 2020. Study participants were randomized into two groups: experimental group, undergoing a 1-month rehabilitative program consisting of MLD and IPC (with a compression of 60 to 80 mmHg); control group, undergoing MLD and a sham IPC (with compression of <30 mmHg). The primary outcome was the lower limb lymphedema reduction, assessed by the circumferential method (CM). Secondary outcomes were: passive range of motion (pROM) of hip, knee, and ankle; quality of life; laboratory exams as fasting plasma glucose and HbA1c. At baseline (T0) and at the end of the 1-month rehabilitative treatment (T1), all the outcome measures were assessed, except for the Hb1Ac evaluated after three months. *Results*: Out of 66 T2DM patients recruited, only 30 respected the eligibility criteria and were randomly allocated into 2 groups: experimental group (n = 15; mean age: 54.2 ± 4.9 years) and control group (n = 15; mean age: 54.0 ± 5.5 years). At the intra-group analysis, the experimental group showed a statistically significant improvement of all outcome measures (*p* < 0.05). The between-group analysis showed a statistically significant improvement in pROM of the hip, knee, ankle, EQ-VAS, and EQ5D3L index at T1. *Conclusions*: A multimodal approach consisting of IPC and MLD showed to play a role in reducing lower limb lymphedema, with an increase of pROM and HRQoL. Since these are preliminary data, further studies are needed.

## 1. Introduction

Type 2 diabetes mellitus (T2DM) is a metabolic disease characterized by insulin secretion reduction or insulin resistance that results in hyperglycemia [1]. This pathologic condition of altered glycemic levels is associated with chronic damage, dysfunction, and failure of different organs, such as eyes, kidneys, myelin sheath, heart, and blood vessels [2]. In 2015, according to the data of the International Diabetes Federation [3], people affected by diabetes worldwide were about 415 million, and this number is expected to increase over the next few years. Moreover, about half of patients with diabetes are not diagnosed and are more prone to complications [4]. Several studies demonstrated a strong correlation among T2DM, obesity, high levels of circulating lipoproteins, and lymphatic dysfunction [5,6,7]. In particular, the lymphatic vasculature seems to be implicated in a variety of lipid-related pathologies [8]. In 2014, Savetsky et al. stated that obese patients presented a higher risk for lymphedema resulting from an impaired baseline lymphatic clearance and an increased propensity for inflammation in response to injury [9].

T2DM microvascular-related complications include neuropathy, nephropathy, and retinopathy; macrovascular complications comprise cardiovascular disease, stroke, and peripheral artery disease (PAD) [10]. A typical symptom of diabetic patients is diabetic foot syndrome, a leading cause of lower limb amputation, defined as a foot ulcer associated with neuropathy, PAD, and infection [4]. Microvascular alterations and cellular dysfunctions occur both in clinical and experimental diabetes mellitus. Impairments due to diabetes cause a venular permeability increase and reduce the early cellular reaction in the inflammatory lesion; this induces overexpression of vascular endothelial growth factor (VEGF). However, while authors thoroughly studied the diabetic alterations of vascular blood beds, abnormalities in the lymphatic system under this condition are not yet well known [11]. The lymphatic system’s role is homeostasis maintenance thanks to fluid recirculation [12]. Pathological conditions may alter the lymphatic system cellular content, resulting in inefficient transport and filtration of fluid, macromolecules, or cells from the interstitium [11].

Lymphedema is the effect of chronic fluid accumulation in the interstitial tissue resulting in impaired physical function, recurrent infections, ulcerations, pain, limb numbness, heaviness, and tightness, with a health-related quality of life (HRQoL) deterioration because of limb swelling [13]. This chronic condition most commonly occurs in the lower extremities (legs and feet) [14]. The precise arm volume measurement allows the early detection of the interstitial accumulation of liquid and could reduce the incidence of irreversible stages [15]. Several methods were proposed in the literature to evaluate and monitor the upper and lower limbs lymphedema. Nowadays, the gold standard for the limb volume measurement is the immersion of it in a tank of water and the diagnosis could be made in the presence of a displacement of 10% in volume or 200mL of fluid [16,17]. This method is now considered imprecise, complex, time-consuming, and contraindicated in the case of skin lesions and infections, especially in the COVID-19 era [18]. Given that the truncated cone solid can be considered as a proxy of the arm shape, the measurement of circumferences across the limb could be used to estimate its total volume [18,19]. Objective evaluation of lower limb lymphedema consists in comparing the circumferential evaluation of the involved limb with the opposite one [20]. A circumferential difference of 2 to 2.5 cm on several levels between the two legs seems consensual to make a diagnosis of lymphedema, even if a unique definition is lacking [21]. This method is affected by heterogeneity due to different operators performing the measurement [22]. While a promising tool that has recently emerged for arm volume measurement is a three-dimensional laser scanner (3DLS) [23,24,25]. This device allows the real-time digital reconstruction of three-dimensional objects and shows great performances in terms of accuracy, non-invasiveness, and cost-effectiveness [26]. Moreover, the 3DLS detects extremely small variations of volume including the presence and reduction of gibbousness and swelling (e.g., a consequence of bandages, press therapy, manual lymph drainage) [25]. The lymphoscintigraphy provides an objective measurement and adequately characterizes the severity of lymphedema [15]. In fact, this imaging technique allows the regional lymph nodes visualization and deep lymphatic nodes localization; nevertheless, this approach is difficult to be applied in the common clinical practice [15]. The indocyanine green (ICG) lymphangiography is a newly developed diagnostic approach that consents to delineating the lymphatics vessels, without any radioactive exposure [26].

Therapeutically, there is no consensus on the appropriate treatment of lymphedema [27,28]. However, the lymphatic system function improvement to help extracellular fluid absorption is possible thanks to complex decongestive therapy (CDT). This approach is one of the most effective therapeutic interventions and consists of patient education on skin hygiene, manual lymphatic drainage (MLD), multilayered bandaging (MLB) treatment, short-stretch compression, and exercise with the bandage [29,30,31,32]. MLD is a massage technique that, providing specific movements in several parts of the body with different pressure, causes muscle relaxation and allows physiological blood and lymphatic circulation, resulting in the renovation of extracellular biological fluids and excretion of catabolic products [33]. Inelastic compression (IC) is a novel approach based on the self-adaption and self-application of IC wraps/garments that allow for increased patient compliance with the treatment. This tool was proposed as a therapeutic alternative in combination with MLD in the clinical rehabilitative management of lymphedema [24]. Intermittent pneumatic compression (IPC) involves a device that allowed the administration of gradual pressure gradients on lymph vessels and improves the lymph flow [34]. This tool could apply pressures between 0 and 300 mmHg pressure, but pressures between 30 and 60 mmHg were generally preferred. Mukherjee et al., in 2021, firstly provided evidence that the in vivo modulation of lymphatic contractility by external oscillatory pressure depends on the parameters of the stimulation [35]. Specifically, IPC modulates function depending on the oscillatory pressure wave frequency and propagation speed [35,36]. In rat models, the oscillatory shear stress induced a lymphangion contractility adaptation as a function of its intrinsic contractility and shear sensitivity [35,36].

In 2012, Palazzin et al. reported that MLD of lower limbs led to a reduction in blood and urinary glucose in patients affected by type 1 diabetes mellitus [37]. Recently, in 2021, Thorn et al. [38] demonstrated that IPC of an upper limb induced arterial inflow vessels vasodilation with enhanced perfusion and determined the maximum capillary recruitment in subjects suffering from T2DM. Moreover, Nikolovska et al. showed that this tool speeds up lymphatic drainage and disposal of metabolic products [39]. However, to date, there is no evidence of the efficacy of the combination of MLD and IPC in modifying glycemic parameters in patients with lymphedema due to diabetes.

Therefore, by the present pilot randomized controlled trial (RCT) we sought to evaluate the efficacy of a synergistic bilateral treatment with MLD and IPC in reducing lower limb lymphedema and, as secondary outcome, the glycemic parameters, in patients affected by T2DM.

## 2. Materials and Methods

### 2.1. Participants

We recruited patients from the Outpatient Service of the Department of Internal Medicine, University Hospital “Mater Domini”, Catanzaro, Italy in a 6-month period, lasting from July to December 2020. The inclusion criteria were: (1) clinical diagnosis of T2DM treated with injective or oral hypoglycemic treatment not been modified in the previous 3 months; (2) diagnosis of lower limb lymphedema (stage II–IV); (3) adult people (age > 18 years old); (4) body mass index from 20 to 30 kg/m^2^; (5) not taking medications able to modify glucose metabolism (e.g., cortisone, diuretics, antipsychotics, transcriptase or protease inhibitors) in the previous 3 months; (6) no significant changes in dietary regimen in the previous 3 months; (7) absence of trauma and/or other conditions able to modify limb structure and volume; (8) signature of a freely given and informed consent. The exclusion criteria were: (1) scars, dermatitis, or hematomas on lower limbs; (2) other comorbidities such as hypertension, dyslipidemia, and thyroid disorder; (3) contraindications to the use of IPC, such as neoplasia, infective and/or inflammatory process, renal or cardiac or pulmonary edema, deep venous thrombosis, fever, pregnancy, chronic obstructive pulmonary disease, and hypotension). This pilot RCT was performed in accordance with the CONSORT Guidelines [40] and approved by the Ethical Committee of the University of Catanzaro “Magna Graecia” (number: 100/16 April 2020). All the participants were asked to carefully read and sign an informed consent before their enrollment, and researchers provided it to protect their privacy and the study procedures according to the Declaration of Helsinki.

### 2.2. Intervention

The enrolled patients were randomly allocated into 2 groups: experimental group, undergoing 3 sessions per week for 4 weeks, consisting of 30 min of bilateral MLD with the Leduc method and 60 min of bilateral IPC through a specific 12-chamber device (BTL-6000 Lymphastim©, BTL, Salerno, Italy) with compression of 60 to 80 mmHg (see Figure 1); control group, undergoing 3 sessions per week for 4 weeks, consisting of 30 min of bilateral MLD with the Leduc method and 60 min of bilateral IPC through a specific 12-chambers device (BTL-6000 Lymphastim©, BTL, Salerno, Italy) with a compression of <30 mmHg (sham treatment). Participants were monitored closely for possible adverse events. 

### 2.3. Outcome Measures

We considered, as a primary outcome, the reduction of lower limb lymphedema, assessed by the circumferential method (CM) of the most involved limb. The CM refers to the assumption that the arm shape is a proxy of a truncated cone solid, requiring the measurement of specific circumferences across the arm to infer its volume. Several studies questioned the sensitivity of CM due to the gibbousness of the upper limbs, but this method is still largely adopted in clinical practice [24]. Secondary outcomes were a passive range of motion (pROM) of hip flexion, knee flexion, and ankle dorsiflexion of the most involved limb; HRQoL, using EuroQol 5 Dimensions 3 Levels Index (EQ5D3L Index), and EuroQol visual analogue scale (EQ-VAS); laboratory exams, as fasting plasma glucose, assayed using enzymatic methods (Roche Diagnostics, Mannheim, Germany), and HbA1c.

At the baseline (T0) and at the end of the one-month rehabilitative treatment (T1), all the outcome measures were assessed, except for the Hb1Ac evaluated after three months. Indeed, this latter represents an approved marker for long-term glycemic control because its lifespan is estimated to be about 90–120 days, thus a blood sample (3–4 mL) was collected at the baseline and after three months. HbA1c was measured with high-performance liquid chromatography using an NGSP-certified automated analyzer (Adams HA-8160 HbA1c analyzer; Menarini, Florence, Italy).

### 2.4. Statistical Analysis

According to the Shapiro–Wilk test, a non-normal distribution of the data was found. We evaluated the differences between groups at the baseline by the Mann-Whitney U test for both demographic and outcome measures. Friedman’s test for repeated measures with Dunn’s multiple comparison test was used to analyze differences between single-variable measurements in each group at T0 and T1 (intra-group analysis). The Kruskal-Wallis test assessed differences between groups (inter-group analysis) for the primary outcomes. Thus, we studied differences between groups for secondary outcome variables at T1 with the Kruskal-Wallis test. Finally, we measured the differences in effect sizes for non-parametric distributions as a biserial rank correlation. Higher coefficients denote a greater entity of the relationship between the variables; as positive correlations denote a relationship traveling on the same trajectory as negative correlations denote a relationship traveling in different directions. A *p*-value < 0.05 was considered statistically significant. Statistical analysis was performed with STATA 16.1 Edition (StataCorp LLC., College Station, TX, USA).

## 3. Results

We recruited 63 patients, where 33 were excluded (25 did not meet inclusion criteria, 6 declined to participate, and 2 refused because of the worsening of other pathologies). Thus, we enrolled 30 patients that were randomly allocated by software into 2 groups of 15 subjects each, with a 1:1 distribution and no blocks, as proposed by Dobkin et al. [41] for pilot studies (Figure 2 describes the study flow chart).

Demographic data and outcome measures at baseline were reported as means ± standard deviations and were not different between groups (see Table 1 for further details).

The experimental group consisted of 9 males and 6 females, mean aged 54.2 ± 4.9 years, with a mean time from the diagnosis of 2.8 ± 2.6 years; the control group consisted of 15 subjects, 10 males and 5 females, aged 54.0 ± 5.5 years, with a mean time from the diagnosis of 2.6 ± 2.3 years. None of the participants changed their therapy or diet, neither assuming drugs that might modify the glucose metabolism during the study period. Five patients (33.3%) in the experimental group and 3 patients (20.0%) in the control group referred urinary urgency after one IPC session, whereas a single subject (6.7%) in the experimental group had just one hypotensive episode during the rehabilitative treatment. Participants did not report major adverse events during the study period and there were no dropouts. Figure 3 depicts the results in terms of CM assessments.

At the intra-group analysis, the experimental group showed a statistically significant improvement of all outcome measures (*p* < 0.05). The control group presented a statistically significant improvement (*p* < 0.05) in HbA1c, pROM of the hip, knee, ankle, EQ-VAS, and EQ-5D. At the inter-group analysis, pROM of the hip, knee, ankle, EQ-VAS, and EQ5D3L Index presented a statistically significant difference at T1. The between-group analysis showed a statistically significant improvement (*p* < 0.05) in pROM of the hip, knee, ankle, EQ-VAS, and EQ5D3L index at T1. Results were described in Table 2.

Considering that the control group showed significant improvements as the experimental group, we performed a comparison of the effect size of the differences via the correlation biserial rank, reporting a negative direction of correlation and showing that the differences in the experimental group were greater (see Table 3 for more details).

## 4. Discussion

The aim of this pilot study was to evaluate the lymphatic drainage role through IPC on glycated hemoglobin and its impact on the quality of life in subjects affected by T2DM. We reported an increase in lower limb range of motion and perceived quality of life improvement. Furthermore, both the experimental and the control group showed a statistically significant reduction in HbA1c. HbA1c was the primary tool for assessing glycemic control and has a strong predictive value for diabetes complications [23,25]. Achieving HbA1c targets of <7% was shown to reduce microvascular complications of T2DM when instituted early in the course of disease [18,25]. A further lowering of A1C from 7% to 6% is associated with a further reduction in the risk of microvascular complications. Our pilot RCT shows that the combination of IPC and MLD might be effective in reduction of lymphedema. However, we did not find a statistically significant difference between groups in terms of HbA1c (*p* = 0.518). Thus, it should be noted that MLD still represents the recommended treatment for the decrease of glycemic parameters in patients affected by lymphedema and T2DM.

Lymphedema is a manageable condition, although hard to treat. Due to this, in recent years, the focus was on avoiding its progression and complications. The most used approach is MLD, an effective and safe method; however, this requires the employment of a properly trained therapist for many hours per day. Hence, the use of IPC was proposed [42]. Authors extensively analyzed IPC for patients undergoing mastectomy for breast cancer [43,44], but its use in the lymphedema of patients affected by diabetes mellitus was lesser studied. IPC applied on lower limbs allows the transmission of the circumferential pressure to the subcutaneous tissue and muscles [45]. If the interstitial pressure in the extracellular space exceeds the hydrostatic pressure, the fluids accumulated in the third space are brought back into the venous circulation. Thus, a decrease of the cutaneous and subcutaneous tissues tensile stretch and a reduction in the limb volume were observed [46]. This mechanism might justify the pROM improvement of lower limb joints, especially concerning knee flexion, which can be hindered in the last degrees by the presence of edema and associated pain. Rockson et al., in 2011, analyzed its possible side effects, such as genital edema and a fibrous tissue ring over the device sleeve proximal edge, and demonstrated that IPC did not appear to have any significant adverse effects [47]. Problems related to venous return and cardiac overload arose with the old IPC devices. These tools consisted of single or double chambers with the same number of channels, with a worse compression adjustment. It is crucial to select the correct parameters for IPC that might enhance overall lymphatic transport by modulating the lymphatic contractility, considering that the increase in mechanosensitivity is an acute response to lymphatic injury, leading to an increase in the pump-like behavior of lymphatics [35,48]. Furthermore, the use of high pressures (>80 mmHg) favored excessive blood movement [49]. In our study, we applied pressure between 60 and 80 mmHg to protect patients from hemodynamic overload. The maintenance of arterial pressure before and after the treatment session confirms the stability of the cardiovascular system.

We also considered the impact on the quality of life of those treated with IPC. Lymphedema results in an inability to carry out the work and activities of daily life. In this context, EQ-5D is considered to have validity, reliability, and responsiveness in T2DM and could be useful for modeling health outcomes in economic evaluations of health program [50]. Participants in the intervention group showed statistically significant improvement after IPC therapy. In the end, the method is accessible and low cost [51]. Moreover, it can be standardized: devices allow pressure regulation, and it is not operator-dependent [49]. IPC is a safe method and may also be administered to patients suffering from heart failure at pressures that do not affect cardiovascular stability [52].

This study has several limitations: first, the small sample size, even though it is a single-center study; second, the scarce literature on this topic did not allow the definition of specific outcomes for assessing any intervention on lymphedema in T2DM; third, the lack of data on glycosylated albumin that could be useful as an indicator of glycemic control.

## 5. Conclusions

To our knowledge, this pilot study is the first to investigate IPC’s role associated with MLD in T2DM lymphedema patients. IPC could facilitate fluid drainage in interstitial tissues of the lower limbs, resulting in a range of motion increase. Moreover, it might improve the perception of the quality of life. Further studies are necessary to define the action of IPC on this pathology, considering the small sample.

## Figures and Tables

**Figure 1 medicina-57-01018-f001:**
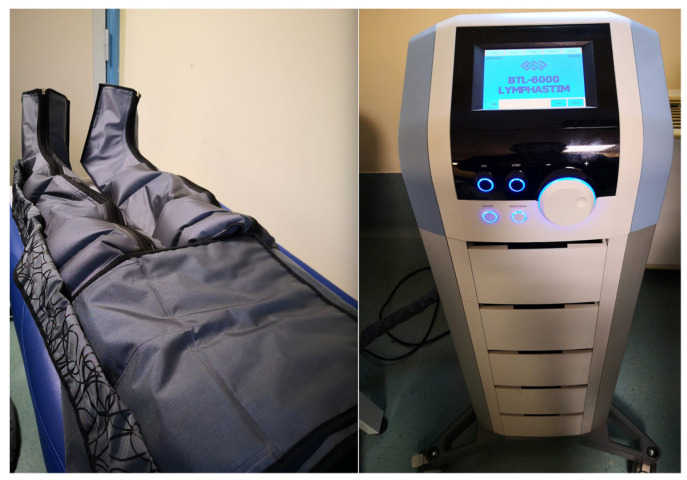
Intermittent pneumatic compression.

**Figure 2 medicina-57-01018-f002:**
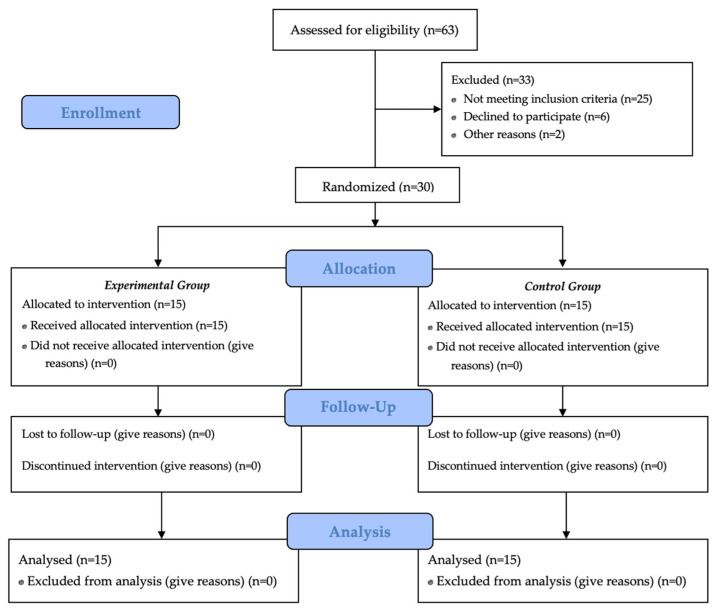
Study flow chart.

**Figure 3 medicina-57-01018-f003:**
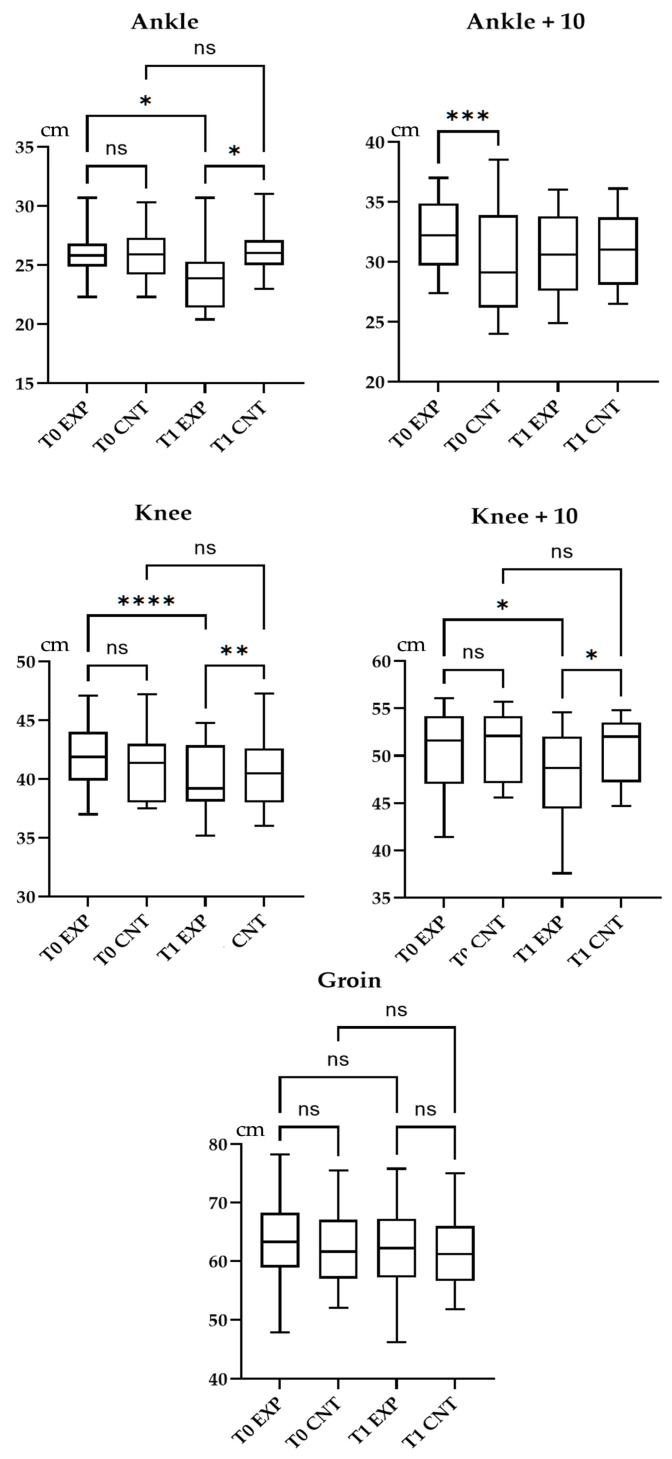
Box and whisker plot of intra-group and inter-group analysis of passive range of motion of the hip, knee, and ankle.

**Table 1 medicina-57-01018-t001:** Baseline demographic and clinical characteristics of study population (n = 30).

	Total (n = 30)	Experimental Group (n = 15)	Control Group (n = 15)
Sex (male/female)	19/11	9/6	10/5
Age (years)	54.1 (5.1)	54.2 (4.9)	54.0 (5.5)
BMI (kg/m^2^)	26.6 (3.4)	27.0 (3.6)	26.2 (3.2)
Time from diagnosis	2.7 (2.4)	2.8 (2.6)	2.6 (2.3)
Type of treatment (injective/oral)	5/25	2/13	3/12

Continuous variables are expressed as means (standard deviations); ranges are expressed as x/y. Abbreviations: BMI: body mass index.

**Table 2 medicina-57-01018-t002:** Intra and inter-group differences in outcomes measures in the study population.

	Experimental Group T0	Experimental Group T1	*p* Value	Control Group T0	Control Group T1	*p* Value	Between-Group Difference at T1 *p* Value
Ankle CM	26.0 (2.5)	23.9 (2.7)	<0.001	25.9 (2.1)	25.2 (1.9)	<0.001	0.152
Ankle + 10 cm CM	31.9 (3.0)	30.4 (3.5)	0.006	31.8 (3.1)	31.1 (3.2)	0.003	0.001
Knee CM	42.0 (2.9)	39.8 (3.0)	<0.001	41.9 (3.1)	40.4 (3.6)	<0.001	0.128
Knee + 10 cm CM	50.6 (4.4)	48.9 (4.4)	<0.001	51.4 (3.6)	50.6 (3.4)	<0.001	0.103
Groin CM	63.3 (7.1)	62.0 (7.0)	<0.001	62.7 (6.9)	62.0 (6.5)	0.023	0.990
Hip flexion pROM (degrees)	111.0 (5.2)	117.5 (3.8)	<0.001	110.9 (6.3)	114.1 (5.4)	<0.001	0.010
Knee flexion pROM (degrees)	110.5 (7.4)	116.3 (4.7)	0.001	110.3 (7.5)	112.1 (6.1)	0.004	0.003
Ankle dorsiflexion pROM (degrees)	12.7 (3.7)	16.3 (2.7)	0.032	12.9 (3.2)	14.6 (3.6)	<0.001	0.014
EQ5D3L Index	0.4 (0.3)	0.2 (0.2)	0.007	0.5 (0.3)	0.4 (0.3)	0.014	<0.001
EQ-VAS	88.5 (10.7)	92.8 (9.0)	0.006	88.9 (10.6)	91.0 (9.7)	0.002	0.022
Fasting blood glucose (mg/dL)	124.8 (18.0)	114.4 (17.2)	0.002	123.8 (10.0)	122.8 (11.4)	0.424	0.199
Hb1Ac (mg/dL)	6.8 (1.0)	6.5 (1.0)	0.018	6.6 (1.2)	6.3 (1.0)	<0.001	0.518

Data are presented as means (standard deviations). Abbreviations: CM = circumferential method; EQ5D3L: EuroQol 5 Dimensions 3 Levels; EQ-VAS: EuroQol visual analogue scale; Hb1Ac = glycated hemoglobin; pROM = passive range of motion.

**Table 3 medicina-57-01018-t003:** Rank biserial correlation between experimental and control group in the outcome measures differences.

		*p* Value	r [95%CI]	*p* Value
Ankle CM	Exp vs. Cnt	0.030	−0.73 [−0.91; −0.35]	<0.001
Ankle +10 cm CM	Exp vs. Cnt	0.128	−0.45 [−0.63; 0.10]	0.495
Knee CM	Exp vs. Cnt	0.002	−0.83 [−0.94; −0.25]	0.003
Knee +10 cm CM	Exp vs. Cnt	0.041	−0.73 [−0.91; −0.35]	<0.001
Groin CM	Exp vs. Cnt	0.999	−0.33 [−0.49; 0.09]	0.625
Hip flexion pROM (degrees)	Exp vs. Cnt	0.010	−0.65 [−0.81; −0.25]	0.005
Knee flexion pROM (degrees)	Exp vs. Cnt	0.003	−0.70 [−0.82; −0.21]	0.006
Ankle dorsiflexion pROM (degrees)	Exp vs. Cnt	0.014	−0.44 [−0.69; −0.23]	0.041
EQ5D3L Index	Exp vs. Cnt	<0.001	−0.62 [−0.82; −0.43]	<0.001
EQ-VAS	Exp vs. Cnt	0.022	−0.65 [−0.87; −0.26]	0.011
Fasting blood glucose (mg/dL)	Exp vs. Cnt	0.199	−0.42 [−0.63; 0.20]	0.806
Hb1Ac (mg/dL)	Exp vs. Cnt	0.518	−0.12 [−0.23; 0.24]	1.351

Data are presented as biserial rank correlation [confidence interval]. Abbreviations: CM = circumferential method; Cnt = control; EQ5D3L = EuroQol 5 Dimensions 3 Levels; EQ-VAS = EuroQol visual analogue scale; Exp = Experimental; Hb1Ac = glycated hemoglobin; pROM = passive range of motion.

## Data Availability

Dataset is available on request.

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
