# Peer review of "Effects of Intermittent Pneumatic Compression on Lower Limb Lymphedema in Patients with Type 2 Diabetes Mellitus: A Pilot Randomized Controlled Trial"

_medicina, 2021, doi:10.3390/medicina57101018_

Round 1
Reviewer 1 Report
I cheer the authors for the interesting paper they wrote. Conservative lymphedema management represents the firs-line treatment nowadays, any improvement in this field may led to better disease's control and patients' quality of life improvement. The manuscript appears consistent with the leading theme of the current special issue of the journal and concepts are shown clearly and interestingly. Nevertheless, English grammar should be revised and minor spelling errors or repetitions of terms should be corrected. Moreover, in the Introduction section, the authors state: "the gold standard for the limb volume measure-76 ment is the immersion of it in a tank of water and the diagnosis could be made in the 77 presence of a displacement of 10% in volume or 200mL of fluid". That is interesting, but other more objective lymphedema assessment methods should be described here, as traditional lymphangiography or ICG-lymphangiography.
Author Response
I cheer the authors for the interesting paper they wrote. Conservative lymphedema management represents the firs-line treatment nowadays, any improvement in this field may led to better disease's control and patients' quality of life improvement. The manuscript appears consistent with the leading theme of the current special issue of the journal and concepts are shown clearly and interestingly.
We are glad that Reviewer 1 has appreciated our work. We agree that the knowledge on rehabilitative management of lymphedema should be improved.
Nevertheless, English grammar should be revised and minor spelling errors or repetitions of terms should be corrected.
We reviewed the English grammar and fixed minor spelling errors.
Moreover, in the Introduction section, the authors state: "the gold standard for the limb volume measure-76 ment is the immersion of it in a tank of water and the diagnosis could be made in the 77 presence of a displacement of 10% in volume or 200mL of fluid". That is interesting, but other more objective lymphedema assessment methods should be described here, as traditional lymphangiography or ICG-lymphangiography
We would like to thank you for the insightful comment. We improved the Introduction Section, as suggested.
Reviewer 2 Report
General comments:
The authors present a study where the importance of intermittent pneumatic compression (IPC) is considered in the presence of manual lymphatic drainage (MLD) for the treatment of lymphedema due to Type-2 Diabetes Mellitus (T2DM). This is a new study that expands upon our understanding of the effects of IPC on lymphedema management and also tries to investigate if IPC intervention can improve glycemic indexes. The study is well designed, with thoughtful inclusion and exclusion criteria, and a wide range of outcome measures are studied. However, I think that the introduction and discussion sections need to be organized better. These sections can be broken up into multiple paragraphs, with each paragraph focusing on a certain aspect or conclusion. I also have several comments about data representation and interpretation that need to be addressed before the paper is ready to be accepted.
Major comments:
- Can the authors explain why they chose an IPC with reduced pressure as a control instead of not having a no-IPC control group? The effect of IPC amplitude on lymphatic function has not been well characterized (to my knowledge). A no-IPC control group would have made for a stronger comparison.
- The authors draw attention to data in a way that seems to support their hypotheses. E.g. Hb1AC values were highlighted to be significantly reduced in the experimental group, but the same is also true for the control group. In fact, Hb1AC was not significantly different between experimental and control groups, which seems to contradict the hypothesis that IPC improves glycemic levels in T2DM patients. The authors should rewrite the discussion explaining these negative results as well, and clearly conclude that IPC did not change glycemic levels as per the outcomes they studied.
- Not all the outcome measures have been presented in a graphical format, which will be helpful in assessing the data.
- I will urge the authors to show the individual datapoints in the graphs and, if possible, indicate the repeated measures by connected lines. Some of the intragroup and intergroup comparisons have an extremely low p value, despite having high variances and similar means. Having the datapoints in the graph, along with an indication of which data points are paired, will help viewers assess the validity of the statistical tests.
- While the authors focus on the intergroup and intragroup differences, it is interesting to note that the control group showed significant improvements across almost all the metrics. A comparison of the effect size of the differences (e.g. using Cohen’s D) will help support the fact that the differences in the experimental group were larger.
- In lines 274-275, when discussing the parameters for IPC stimulation, please discuss Mukherjee et al. 2021 (Journal of Physiology) since it describes the importance of parameter selection for IPC on lymphatic function. Choosing the right parameters for IPC can indeed benefit the patients and should be investigated in future studies.
Minor comments:
- Several minor and major grammatical errors are present need to be corrected. I highly recommend the authors to proofread the manuscript carefully before a resubmission. The edits I have listed below may not be comprehensive.
- Line 48 – please specify what you mean by action reduction
- Line 54 – “…are more apt to the typical complications of disease development” should be reworded. I assume the authors meant that the patients are unaware of the symptoms of diabetes and ignore them as symptoms of other, more benign, conditions.
- Line 58 – “…typical pattern” should be changed to “typical symptom”
- Line 62 – “Impairs in” should be changed to “impairments due to”
- Line 65-66 – Reword this sentence. Fluid excretion should be changed to fluid recirculation.
- Line 72 – “cronic”
- Line 72 – “occurs in lower limbs”. A more accurate term might be extremities (such as arms and legs).
- Line 73 – “to early detect” to “early detection of”
- Line 167 – “as” to “such as”
- Please be consistent with writing Hb1AC (written as HB1AC in some places)
- Line 55 – “Obesity can heighten the risk of lymphedema.” Relevant literature should be cited here showing this link. The authors should also the consider the angle of reduced lymphatic function related to obesity and how it can connect to lymphedema. Some relevant studies should be cited including, but not limited to, Kassis et al. 2016 (AJP-Gastrointestinal and Liver Physiology), Blum et al. 2014 (PLOS One), Savetsky et al. 2014 (AJP Heart Circ).
- Line 89 – It is confusing to introduce bioimpedance in the same sentence as 3DLS. Please restrict the discussion only to 3DLS in this part.
- Line 114-115 – It will be prudent to discuss the recent literature on how oscillatory forces modulate lymphatic function in this section – Mukherjee et al. 2021 (Journal of Physiology) is relevant here, since modulation of lymphatic function by IPC is a relatively newly studied area that can have relevance to fluid clearance in lymphedema. A discussion on lymphatic contractility modulation by oscillatory forces will also be beneficial (relevant literature includes Mukherjee et al. 2019 (Scientific Reports), Kornuta et al. 2015 (AJP – Regul. Integr. Comp. Physiol.).
- The authors should clarify what the main goal of the paper is towards the end of the introduction section. Lines 116-118 say that the authors will look at how glycemic parameters change in response to IPC, but lines 119-121 say that the work will look at the reduction of lymphedema due to T2DM. These two questions can coexist but should be explained better.
- It might be an error in Fig. 2 where an n=15 is shown for patients that did not receive allocated intervention. I assume that all 15 patients in the experimental (and control) groups received the intervention (sham or otherwise).
- Figure 3 is missing Y-axis titles. Also, it is not clear what data Figure 3 is representing. The authors should clarify this further in the figure legend.
Author Response
The authors present a study where the importance of intermittent pneumatic compression (IPC) is considered in the presence of manual lymphatic drainage (MLD) for the treatment of lymphedema due to Type-2 Diabetes Mellitus (T2DM). This is a new study that expands upon our understanding of the effects of IPC on lymphedema management and also tries to investigate if IPC intervention can improve glycemic indexes. The study is well designed, with thoughtful inclusion and exclusion criteria, and a wide range of outcome measures are studied.
We are glad that Reviewer 2 has appreciated our work. We agree that the knowledge on rehabilitative management of lymphedema should be improved.
However, I think that the introduction and discussion sections need to be organized better. These sections can be broken up into multiple paragraphs, with each paragraph focusing on a certain aspect or conclusion. I also have several comments about data representation and interpretation that need to be addressed before the paper is ready to be accepted.
We would like to thank the reviewer for the insightful comment. We have improved the Introduction and Discussion Sections as suggested.
- Can the authors explain why they chose an IPC with reduced pressure as a control instead of not having a no-IPC control group? The effect of IPC amplitude on lymphatic function has not been well characterized (to my knowledge). A no-IPC control group would have made for a stronger comparison.
Thanks for the insightful comment. We intended to have 2 groups: one experimental group with IPC pressures ranging from 30 to 60mmHg (commonly applied for therapeutic purposes) and on control group undergoing a sham-IPC group with pressures applied for "aesthetic" aims.
- The authors draw attention to data in a way that seems to support their hypotheses. E.g. Hb1AC values were highlighted to be significantly reduced in the experimental group, but the same is also true for the control group. In fact, Hb1AC was not significantly different between experimental and control groups, which seems to contradict the hypothesis that IPC improves glycemic levels in T2DM patients. The authors should rewrite the discussion explaining these negative results as well, and clearly conclude that IPC did not change glycemic levels as per the outcomes they studied.
Thank you for this suggestion. In the Results section we reported that at the intra-group analysis, the experimental group showed a statistically significant improvement of all outcome measures, whereas the control group presented a statistically significant improvement only in HbA1c, pROM of the hip, knee, ankle, EQ-VAS, and EQ-5D. Furthermore, the between-group analysis did not show a statistically significant improvement in Hb1A1c at T1. Thus, we could conclude that combination of IPC and MLD is effective in the reduction of HbA1c. Moreover, we modified the Discussion Section, according to your suggestion.
- Not all the outcome measures have been presented in a graphical format, which will be helpful in assessing the data.
We would like to thank the reviewer for the comment. We have provided in a graphical format only the primary outcome (Figure 3). All the outcome measures are reported in Table 2.
- I will urge the authors to show the individual datapoints in the graphs and, if possible, indicate the repeated measures by connected lines. Some of the intragroup and intergroup comparisons have an extremely low p value, despite having high variances and similar means. Having the datapoints in the graph, along with an indication of which data points are paired, will help viewers assess the validity of the statistical tests.
Thanks for the helpful comment. Being non-parametric distributions, the connection of values with lines would exacerbate the confusion, for this reason we have reformulated the figure by depriving it of points and keeping medians and interquartile ranges
- While the authors focus on the intergroup and intragroup differences, it is interesting to note that the control group showed significant improvements across almost all the metrics. A comparison of the effect size of the differences (e.g. using Cohen’s D) will help support the fact that the differences in the experimental group were larger.
Thanks for the helpful comment. Being non-parametric distributions, we have provided an analysis of biserial correlation ranks, according to Kerby (Kerby DS. The Simple Difference Formula: An Approach to Teaching Nonparametric Correlation. Comprehensive Psychology. January 2014. doi:10.2466/11.IT.3.1). Thus, we have added a newTable 3.
- In lines 274-275, when discussing the parameters for IPC stimulation, please discuss Mukherjee et al. 2021 (Journal of Physiology) since it describes the importance of parameter selection for IPC on lymphatic function. Choosing the right parameters for IPC can indeed benefit the patients and should be investigated in future studies.
Thank you for your suggestion. We are aware of the importance of an adequate choice of parameters. Accordingly, we added the suggested reference.
Several minor and major grammatical errors are present need to be corrected. I highly recommend the authors to proofread the manuscript carefully before a resubmission. The edits I have listed below may not be comprehensive.
We reviewed the English grammar and fixed minor spelling errors.
- Line 48 – please specify what you mean by action reduction
Thank you. We modified the text, accordingly.
- Line 54 – “…are more apt to the typical complications of disease development” should be reworded. I assume the authors meant that the patients are unaware of the symptoms of diabetes and ignore them as symptoms of other, more benign, conditions.
Thank you. We reformulated the sentence, as suggested.
- Line 65-66 – Reword this sentence. Fluid excretion should be changed to fluid recirculation.
Thank you. We modified the text, accordingly.
- Line 55 – “Obesity can heighten the risk of lymphedema.” Relevant literature should be cited here showing this link. The authors should also the consider the angle of reduced lymphatic function related to obesity and how it can connect to lymphedema. Some relevant studies should be cited including, but not limited to, Kassis et al. 2016 (AJP-Gastrointestinal and Liver Physiology), Blum et al. 2014 (PLOS One), Savetsky et al. 2014 (AJP Heart Circ).
Thank you for the opportunity to deepen this aspect. We implemented the text, discussing the suggested references.
- Line 114-115 – It will be prudent to discuss the recent literature on how oscillatory forces modulate lymphatic function in this section – Mukherjee et al. 2021 (Journal of Physiology) is relevant here, since modulation of lymphatic function by IPC is a relatively newly studied area that can have relevance to fluid clearance in lymphedema. A discussion on lymphatic contractility modulation by oscillatory forces will also be beneficial (relevant literature includes Mukherjee et al. 2019 (Scientific Reports), Kornuta et al. 2015 (AJP – Regul. Integr. Comp. Physiol.).
Thank you for the opportunity to deepen this aspect. We implemented the text, discussing the suggested references.
- The authors should clarify what the main goal of the paper is towards the end of the introduction section. Lines 116-118 say that the authors will look at how glycemic parameters change in response to IPC, but lines 119-121 say that the work will look at the reduction of lymphedema due to T2DM. These two questions can coexist but should be explained better.
Thank you for the suggestion. We have better clarified the aim.
- Figure 3 is missing Y-axis titles. Also, it is not clear what data Figure 3 is representing. The authors should clarify this further in the figure legend.
We have provided a graphical format, providing a new Figure 3.
Round 2
Reviewer 2 Report
I would like to thank the authors for adequately responding to the comments, including further analyses of the data and fleshing out the discussions even further with all the recommended citations. However, there are a couple of minor misinterpretations of the literature cited that need to be corrected:
- Line 124-129: Mukherjee et al. 2021 was not the first paper to provide evidence of in vivo modulation of lymphatic contraction by external oscillatory pressure, but the first to provide evidence that the in vivo modulation of lymphatic contractility by external oscillatory pressure depends on the parameters of the stimulation.
- Line 321-325 - IPC does not enhance the mechanosensitivity of lymphatic vessels. Rather, selecting the correct parameters for IPC can enhance overall lymphatic transport by modulating the lymphatic contractility. The increase in mechanosensitivity (leading to an in crease in the pump-like behavior of lymphatics) is seen as an acute response to lymphatic injury (Nelson 2019, Nature Biomed Engg is also another relevant paper to cite here), which enhances the sensitivity of the lymphatics to the IPC derived forces.
Author Response
I would like to thank the authors for adequately responding to the comments, including further analyses of the data and fleshing out the discussions even further with all the recommended citations.
Thank you for the comment. We are glad that the reviewer has appreciated our work.
However, there are a couple of minor misinterpretations of the literature cited that need to be corrected:
- Line 124-129: Mukherjee et al. 2021 was not the first paper to provide evidence of in vivo modulation of lymphatic contraction by external oscillatory pressure, but the first to provide evidence that the in vivo modulation of lymphatic contractility by external oscillatory pressure depends on the parameters of the stimulation.
Thank you for the comment. We have modified the text, accordingly.
- Line 321-325 - IPC does not enhance the mechanosensitivity of lymphatic vessels. Rather, selecting the correct parameters for IPC can enhance overall lymphatic transport by modulating the lymphatic contractility. The increase in mechanosensitivity (leading to an in crease in the pump-like behavior of lymphatics) is seen as an acute response to lymphatic injury (Nelson 2019, Nature Biomed Engg is also another relevant paper to cite here), which enhances the sensitivity of the lymphatics to the IPC derived forces.
Thank you for the comment. We have modified the text, accordingly.